# Functionally distinct and selectively phosphorylated GPCR subpopulations co-exist in a single cell

Ao Shen[1], Madeline Nieves-Cintron[1], Yawen Deng[1,2], Qian Shi[1], Dhrubajyoti Chowdhury[1], Jinyi Qi[3], Johannes W. Hell [1], Manuel F. Navedo [1] & Yang K. Xiang [1,4]

G protein-coupled receptors (GPCRs) transduce pleiotropic intracellular signals in a broad range of physiological responses and disease states. Activated GPCRs can undergo agonist-induced phosphorylation by G protein receptor kinases (GRKs) and second messenger-dependent protein kinases such as protein kinase A (PKA). Here, we characterize spatially segregated subpopulations of $\beta_2$-adrenergic receptor ($\beta_2$AR) undergoing selective phosphorylation by GRKs or PKA in a single cell. GRKs primarily label monomeric $\beta_2$ARs that undergo endocytosis, whereas PKA modifies dimeric $\beta_2$ARs that remain at the cell surface. In hippocampal neurons, PKA-phosphorylated $\beta_2$ARs are enriched in dendrites, whereas GRK-phosphorylated $\beta_2$ARs accumulate in soma, being excluded from dendrites in a neuron maturation-dependent manner. Moreover, we show that PKA-phosphorylated $\beta_2$ARs are necessary to augment the activity of L-type calcium channel. Collectively, these findings provide evidence that functionally distinct subpopulations of this prototypical GPCR exist in a single cell.

[1] Department of Pharmacology, University of California Davis, Davis, CA 95616, USA. [2] The Second Affiliated Hospital, Sun Yat-sen University, 510120 Guangzhou, China. [3] Department of Biomedical Engineering, University of California Davis, Davis, CA 95616, USA. [4] VA Northern California Health Care System, Mather, CA 95655, USA. Correspondence and requests for materials should be addressed to Y.K.X. (email: ykxiang@ucdavis.edu)

A ctivation of G protein-coupled receptors (GPCRs) transduces the canonical G protein-dependent signal as well as noncanonical G protein-independent signals, frequently via β-arrestins[1,2]. In the past decades, it has been appreciated that some ligands can differentially activate a GPCR via a phenomenon known as functional selectivity or biased signaling. Depending on the receptor, different mechanisms have been proposed for biased GPCR signaling, which include ligand efficacy bias, receptor conformational bias, cell type and/or expression level-caused cellular bias[3-5]. One of the universal features of GPCRs is that they undergo agonist-induced phosphorylation by a variety of kinases, which may also allow distinct structural features that favors receptor binding to different signaling partners[6-8]. Molecular and structural details underlying biased agonism need to be further elucidated, especially how a single ligand–receptor pair can selectively transduce different signals in space and time in a single cell.

$\beta_2AR$, a prototypical GPCR, is involved in memory and learning in the central nervous system, and cardiovascular and metabolism regulation in peripheral systems[9,10]. Stimulation of $\beta_2AR$ promotes phosphorylation of serine 355 and 356 at the receptor C-terminal domain by GRKs, contributing to receptor desensitization and endocytosis[11,12]. $\beta_2AR$ also undergoes phosphorylation by PKA at serine 261 and 262 in the third loop and serine 345 and 346 in the C-terminal domain[11,13]. Here we apply super-resolution imaging together with single molecular analysis to probe $\beta_2AR$ subpopulations that undergo phosphorylation by GRKs and PKA after agonist stimulation. Our results show that GRKs and PKA selectively label two distinct subpopulations of $\beta_2AR$ that are spatially segregated on the plasma membrane and undergo distinct membrane trafficking in both fibroblasts and neurons. Moreover, these two subpopulations exert distinct functions in modulating L-type calcium channel (LTCC) activity and neuron excitability.

## Results

**PKA and GRKs target spatially segregated $\beta_2AR$ subpopulations**. In this study, we characterized the subcellular distribution of $\beta_2ARs$ upon agonist-induced phosphorylation by PKA and GRKs. We used two sets of well-characterized phospho-specific antibodies: anti-pS261/262 (monoclonal 2G3 and 2E1) and anti-pS355/356 (monoclonal 10A5, polyclonal 22191R, and 16719R) antibodies[13–16], and here with mutant $\beta_2AR$ lacking either the PKA (PKAmut) or GRK (GRKmut) sites (Supplementary Fig. 1). $\beta_2ARs$ localize on cell membrane at resting state (Fig. 1a). Using super-resolution structured illumination microscopy (SIM), we found that after acute stimulation with the $\beta AR$ agonist isoproterenol (ISO, 30 s or 1 min), both PKA- and GRK-phosphorylated $\beta_2ARs$ are primarily segregated at the plasma membrane (PM) of HEK293 cells (Fig. 1b, top panel; Fig. 1c, d, Pearson's coefficient $0.078 \pm 0.016$ for ISO 30 s and $0.058 \pm 0.015$ for ISO 1 min, mean $\pm$ s.e.m, three independent experiments). Comparably, PKA- and GRK-phosphorylated $\beta_2ARs$ highly co-localize with total $\beta_2AR$ (Fig. 1b, bottom two panels; Fig. 1c, d, Pearson's coefficient $0.671 \pm 0.035$ and $0.510 \pm 0.039$ for ISO 30 s, $0.601 \pm 0.039$ and $0.507 \pm 0.033$ for ISO 1 min, respectively, mean $\pm$ s.e.m, three independent experiments). After prolonging stimulation with ISO for 5 to 10 min, GRK- and PKA-phosphorylated $\beta_2ARs$ display further spatiotemporal segregation: GRK-phosphorylated $\beta_2ARs$ undergo internalization and form puncta inside the cells, whereas PKA-phosphorylated $\beta_2ARs$ stay on the PM (Fig. 2a, b; Supplementary Fig. 2).

The segregation between PKA- and GRK-phosphorylated $\beta_2AR$ was validated biochemically with immuno-isolation of GRK-phosphorylated FLAG-$\beta_2AR$ with anti-pS355/356 antibody. The remaining $\beta_2AR$ was subsequently immuno-isolated with anti-FLAG antibody, GRK-phosphorylated $\beta_2ARs$ and PKA-phosphorylated $\beta_2ARs$ were enriched in first and second immuno-isolations, respectively (Figs. 1e and 2c). We also

applied surface biotinylation-based fractionation to separate PM from intracellular endosome after stimulation with ISO for 10 min. At a minimal dose of 1 nM ISO, $\beta_2AR$ displayed phosphorylation only at the PKA sites, and the phosphorylated receptors remained at the PM. At a saturated dose of 1 $\mu$M ISO, GRK-phosphorylated $\beta_2ARs$ were partitioned in the endosomal fraction, whereas PKA-phosphorylated $\beta_2ARs$ remained in the PM fraction (Fig. 2d). These data demonstrate two subpopulations of $\beta_2AR$ undergoing phosphorylation by PKA and GRKs and displaying distinct spatial distribution in a single cell.

**PKA- and GRK-p$\beta_2ARs$ display distinct oligomeric states**. We applied our recently developed single-molecule pulldown (SiM-Pull) assay to gain structural insight into $\beta_2AR$ subpopulations that are modified by PKA or GRKs (Fig. 3a)[17–19]. We co-expressed FLAG-mYFP-$\beta_2AR$ and FLAG-mCherry-$\beta_2AR$ at a 1:1 ratio in HEK293 cells, treated the cells with ISO, and pulled down the receptors with either anti-FLAG antibody or phospho-specific antibodies in SiMPull (Fig. 3a; Supplementary Fig. 3a). A reference construct with mYFP and mCherry fused into a single protein displayed $58.8\% \pm 1.6\%$ (mean $\pm$ s.d., three independent experiments) overlap between the two proteins in SiMPull (Fig. 3b, c). The incomplete co-localization arises primarily from immature/inactive chromophores as both mYFP and mCherry display about a 75% fluorescent maturation ratio[18,20,21]. PKA-phosphorylated mCherry-$\beta_2AR$ had more than $20.4\% \pm 1.6\%$ (mean $\pm$ s.d., three independent experiments) overlap with mYFP-$\beta_2AR$. In contrast, GRK-phosphorylated $\beta_2AR$ had less than $3.2\% \pm 0.5\%$ (mean $\pm$ s.d., three independent experiments) overlap between the mCherry and mYFP versions of $\beta_2AR$ (Fig. 3b, c). As control, the total $\beta_2AR$ pulled down with anti-FLAG antibody displayed $13.2\% \pm 0.8\%$ (mean $\pm$ s.d., three independent experiments) of overlap between mYFP and mCherry (Fig. 3b, c).

SiMPull can reveal stoichiometry of protein complexes via single-molecule chromophore photobleaching step analysis when proteins are fluorescently labeled at a one-to-one ratio. For example, the photobleaching of a single monomeric YFP (mYFP) is a discrete process; thus the fluorescence intensity of a protein complex with one or several mYFP molecules drops in a stepwise fashion, and the number of steps reveals the number of mYFP-tagged proteins in the complex (Fig. 4a; Supplementary Fig. 3e–g)[18,20,21]. This approach was validated with reference membrane receptors including mYFP-fused monomeric CD86 and mYFP-fused dimeric CD28 (Supplementary Fig. 3b)[22]. We then used this method to directly determine the number of $\beta_2ARs$ in a complex by counting discrete steps of photobleaching of mYFP molecules. Consistent with our previous report[18], SiMPull photobleaching analysis revealed that $\beta_2ARs$ can be found as a mixture of monomers and dimers ($42.7\% \pm 1.5\%$ dimers, mean $\pm$ s.d., 36 independent experiments). The composition of $\beta_2AR$ was not affected by the expression levels of the receptor

**Fig. 1** PKA- and GRK-p$\beta_2ARs$ are spatially segregated on the plasma membrane. **a**, **b** SIM imaging shows total $\beta_2ARs$, and PKA- and GRK-phosphorylated $\beta_2ARs$, which were stained with anti-FLAG, anti-S261/262 (PKA-p$\beta_2AR$), and anti-S355/356 (GRK-p$\beta_2AR$) specific antibodies, respectively, in HEK293 cells expressing FLAG-tagged $\beta_2AR$ before stimulation (**a**) or after 30 s of stimulation with 1 $\mu$M ISO (**b**). Scale bar, 2 $\mu$m. Representative of $n = 5, 4, 19, 11$, and 13 cells, respectively, three independent experiments. **c** The overlap between two different staining from images in **b** was evaluated by Pearson's correlation coefficient ($n = 19, 11$, and 13 cells, respectively). **d** Quantification of two-color overlap was evaluated in HEK293 cells after 1 min of stimulation with 1 $\mu$M ISO using the same method in **b** ($n = 13, 12$, and 11 cells, respectively, three independent experiments). **e** FLAG-$\beta_2ARs$ expressed in HEK293 cells were stimulated with 1 $\mu$M ISO for 30 s, and cell lysates were subjected to immuno-isolation with anti-pS355/356 (GRK-p$\beta_2AR$) specific antibody (1st immunoprecipitate, IP). The remaining $\beta_2ARs$ in the supernatant were isolated with anti-FLAG antibody (2nd IP). Total $\beta_2AR$ were immuno-isolated directly with anti-FLAG antibody (total IP). The $\beta_2AR$ in total IP and sequential IPs were resolved with SDS-PAGE, and probed with anti-FLAG, anti-pS261/262 (PKA-p$\beta_2AR$), and anti-pS355/356 (GRK-p$\beta_2AR$) antibodies, respectively. Representative of three independent experiments. Molecular weight markers (in kDa) are indicated on the left. Error bars denote s.e.m.; multiplicity adjusted $P$ values are computed by one-way ANOVA followed by Tukey's test between indicated groups

(Supplementary Fig. 3c) or concentrations of ISO (Fig. 4b). In the same cell lysates treated with ISO, PKA-phosphorylated $\beta_2$ARs pulled down with anti-pS261/262 antibody were mainly dimers (68.8% ± 2.1% dimers, mean ± s.d., 26 independent experiments, Fig. 4c and Supplementary Fig. 3d). In comparison, GRK-phosphorylated $\beta_2$AR pulled down with anti-pS355/356 antibody displayed a predominant monomeric composition (86.5% ± 2.1% monomers, mean ± s.d., 33 independent experiments, Fig. 4c).

Together, these results suggest that PKA- and GRK-phosphorylated $\beta_2$ARs have different oligomeric assembly states.

**$\beta_2$AR subpopulations are spatially segregated in neurons.** Activation of $\beta$ARs promotes PKA-phosphorylation of ion channels including LTCC and α-amino-3-hydroxy-5-methyl-4-isoxazolepropionic acid receptor (AMPAR) to modulate

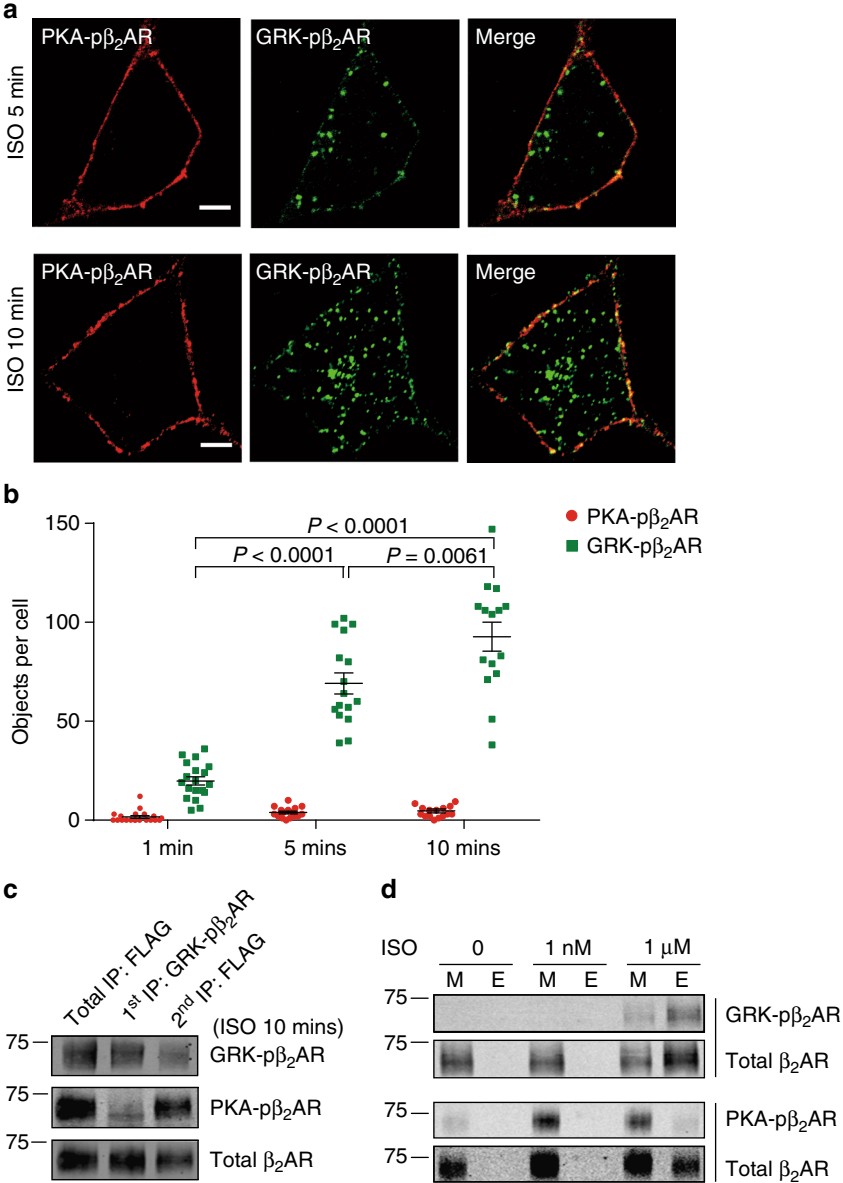

**Fig. 2** PKA- and GRK-p$\beta_2$AR undergo distinct membrane trafficking. **a** FLAG-$\beta_2$ARs expressed in HEK293 cells were stimulated with ISO for indicated times. Confocal imaging shows PKA- and GRK-phosphorylated $\beta_2$ARs, which were stained with anti-pS261/262 (PKA-p$\beta_2$AR) and anti-pS355/356 (GRK-p$\beta_2$AR) antibodies, respectively. Scale bar, 5 μm. Representative of $n = 16$ and 15 cells, respectively, three independent experiments. **b** Numbers of fluorescent objects in each cell in **a** were quantified with ImageJ ($n = 19$, 16, and 15 cells, respectively, three independent experiments). Error bars denote s.e.m.; multiplicity adjusted $P$ values are computed by one-way ANOVA followed by Tukey's test between indicated groups. **c** Immuno-isolation of PKA- and GRK-phosphorylated $\beta_2$ARs in HEK293 cells using same procedure as Fig. 1d after stimulation with 1 μM ISO for 10 min. The $\beta_2$AR in total IP and sequential IPs were resolved in SDS-PAGE, and probed with anti-FLAG, anti-pS261/262 (PKA-p$\beta_2$AR) and anti-pS355/356 (GRK-p$\beta_2$AR) antibodies, respectively. Representative of three independent experiments. **d** FLAG-$\beta_2$ARs expressed in HEK293 cells underwent cell surface biotin labeling and then were stimulated with ISO (1 nM or 1 μM) for 10 min. The biotin-labeled proteins on the plasma membrane were pulled with streptavidin beads, and the leftover biotin-labeled proteins in endosome were isolated by a second precipitation with streptavidin beads. Membrane (M) and endosome (E) fractions were resolved in Western blot with antibodies against FLAG, pS261/262 (PKA-p$\beta_2$AR), and pS355/356 (GRK-p$\beta_2$AR), showing separation of GRK- and PKA-phosphorylated subpopulations of $\beta_2$AR. Representative of three independent experiments. Molecular weight markers (in kDa) are indicated on the left

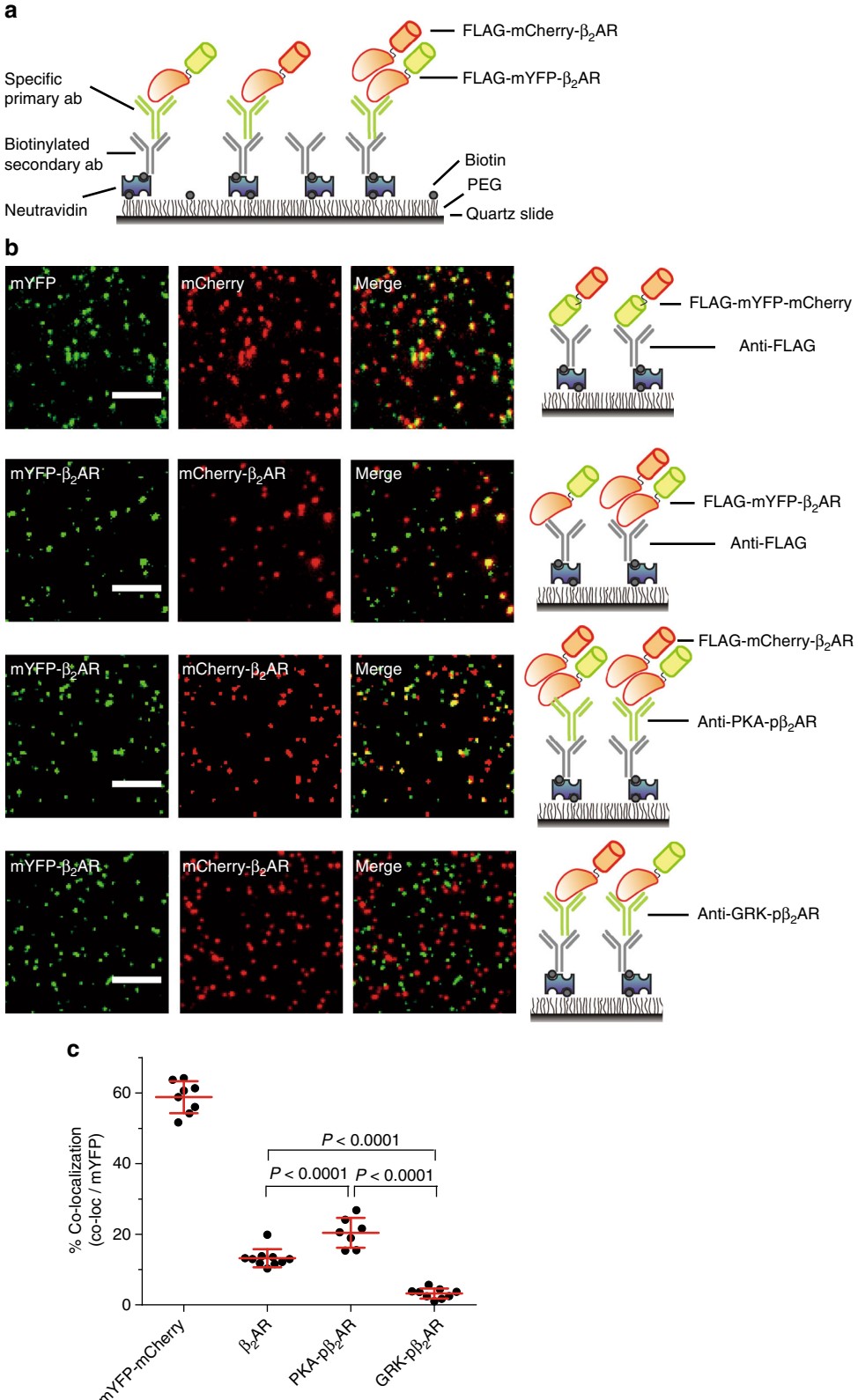

**Fig. 3** PKA- and GRK-p$\beta_2$AR display minimal overlap at the single-molecule level. **a** Schematic of single-molecule pulldown (SiMPull) assay. **b** Representative SiMPull images of a fusion mCherry-mYFP protein, total $\beta_2$AR, PKA-phosphorylated $\beta_2$AR, and GRK-phosphorylated $\beta_2$AR pulled down from cell lysates expressing mCherry-$\beta_2$AR and mYFP-$\beta_2$AR at 1:1 ratio. Scale bar, 5 μm. Representative of $n = 8$, 10, 7, and 9 images, respectively, three independent experiments. **c** Quantification of overlap percentage between mCherry and mYFP from images in **b** ($n = 8$, 10, 7, and 9, respectively). Error bars denote s.d.; multiplicity adjusted $P$ values are computed by one-way ANOVA followed by Tukey's test between indicated groups

membrane potential and synaptic activity[23–25]. In hippocampal immature neurons, while PKA- and GRK-phosphorylated $\beta_2$ARs were found in both soma and dendrites, similar to what had been observed in HEK293 cells, super-resolution images showed lack of co-localization between PKA- and GRK-phosphorylated $\beta_2$ARs (Supplementary Fig. 4). Interestingly, as neurons mature, more GRK-phosphorylated $\beta_2$ARs existed in cell bodies and less is found in distal dendrites (Fig. 5). In contrast, PKA-phosphorylated $\beta_2$ARs were enriched in both proximal and distal dendrites relative to soma in both mature and immature dendrites (Fig. 5; Supplementary Fig. 5). In mature neuron dendrites, GRK-phosphorylated $\beta_2$ARs levels were much lower compared to PKA-phosphorylated $\beta_2$ARs (Fig. 5a, c; Supplementary Fig. 5). These data indicate PKA- and GRK-phosphorylated subpopulations of $\beta_2$AR are in different subcellular compartments in mature neurons and support specific

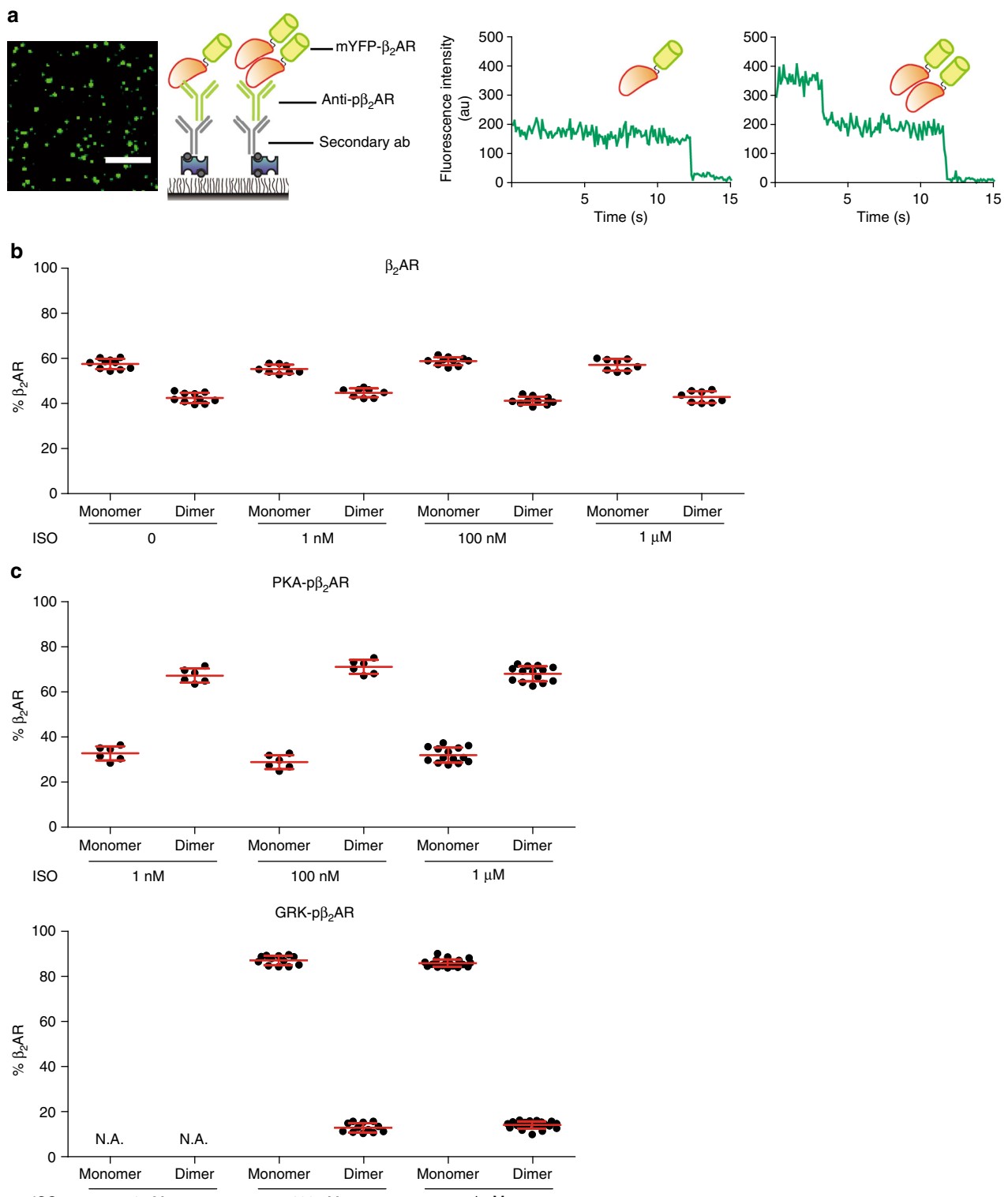

roles of PKA-phosphorylated $\beta_2$ARs in dendritic regions for regulation of ion channels during synaptic transmission.

**PKA-p$\beta_2$ARs control activation of LTCC in hippocampal neurons**. We have previously shown that the LTCC $Ca_v$ $\alpha_1$1.2 subunit forms a membrane complex with $\beta_2$AR in the brain[23], and recently we have reported that ISO induces a $\beta_2$AR-dependent activation of LTCC in neurons, and PKA-phosphorylation of serine 1928 of $\alpha_1$1.2 displaces the $\beta_2$AR from $\alpha_1$1.2 and promotes channel activation[26,27]. Deletion of $\beta_2$AR genes ($\beta_2$AR KO) abolished ISO-induced increases in overall channel activity (nPo) of LTCC $Ca_v$1.2 as measured by single-channel recordings in hippocampal neurons (Fig. 6). In contrast, deletion of $\beta_1$AR gene ($\beta_1$AR KO) did not affect the ISO-induced LTCC responses (Fig. 6). The results suggest that $\beta_2$AR but not $\beta_1$AR play a specific role in modulation of LTCC $Ca_v$1.2 activity in neurons.

We then introduced WT or mutant $\beta_2$ARs lacking either PKA-phosphorylation sites (PKAmut) or GRK-phosphorylation sites (GRKmut) in hippocampal neurons with deficiency of both $\beta_1$AR and $\beta_2$AR genes (DKO) to examine the function of PKA-phosphorylation sites on $\beta_2$AR in activation of LTCC. While WT and $\beta_2$AR-GRKmut promoted PKA-phosphorylation of S1928 on $\alpha_1$1.2 after ISO stimulation, deletion of PKA-phosphorylation sites on $\beta_2$AR abolished ISO-induced PKA-phosphorylation of $\alpha_1$1.2 S1928 (Fig. 7a). Unexpectedly, stimulation of WT and mutant $\beta_2$ARs promoted similar increased in phosphorylation of serine 1700 on $\alpha_1$1.2 in DKO neurons (Fig. 7a). S1700 is another PKA-phosphorylation site on $\alpha_1$1.2 important for upregulation of LTCC activity in heart, but is not relevant in neurons[27–29]. Moreover, reintroduction of WT and $\beta_2$AR-GRKmut promoted ISO-induced dissociation of the receptor from $\alpha_1$1.2 in both DKO hippocampal neurons and HEK293 cells (Fig. 7b; Supplementary Fig. 6), and increases in nPo of LTCC $Ca_v$1.2 in neurons, whereas activation of $\beta_2$AR-PKAmut failed to do so (Fig. 7c, d). Together, these data indicate that ISO-induced PKA-phosphorylation of $\beta_2$AR is necessary to transduce signal to promote PKA-phosphorylation and activation of LTCC $Ca_v$1.2 in hippocampal neurons.

## Discussion

This study reveals that a GPCR ($\beta_2$AR) can be present in functionally distinct subpopulations that are distributed at different subcellular locations in a single cell. These $\beta_2$AR subpopulations undergo agonist-induced phosphorylation by GRKs and second messenger-dependent PKA, respectively, in which GRK- and PKA-phosphorylated subpopulations are segregated into distinct microdomains on the plasma membrane. Moreover, GRK- and PKA-phosphorylated $\beta_2$AR subpopulations display distinct membrane trafficking in both fibroblasts and hippocampal neurons. While GRK-phosphorylated $\beta_2$ARs undergo endocytosis, PKA-phosphorylated $\beta_2$ARs remain on the cell surface. This is consistent with the literature that GRKs play a dominant and necessary role in agonist-induced $\beta_2$AR endocytosis[7,30,31].

In hippocampal neurons, there is a further segregation of GRK- and PKA-phosphorylated $\beta_2$AR subpopulations in a neuron maturation-dependent manner. In immature neurons with 6–8 days of culture in vitro, there is a small but appreciable enrichment of PKA-phosphorylated $\beta_2$ARs in dendrites whereas GRK-phosphorylated $\beta_2$ARs are relatively enriched in soma. In mature neurons with 18–21 days of culture in vitro, GRK-phosphorylated $\beta_2$ARs are almost entirely enriched in soma and excluded from dendrites. In contrast, PKA-phosphorylated $\beta_2$ARs are further enriched in proximal and distal dendrites. It is known that many proteins including sodium and potassium ion channels develop specific functions in neurons due to their selective targeting and distributions on axon, dendrites, and soma in a single neuron[32,33]. To our knowledge, our data show that a GPCR, based on the subcellular distribution, can exist in different functional subpopulations in a fully mature neuron, which offers a distinct mechanism to develop functional heterogeneity of a protein in highly differentiated neurons. The molecular mechanisms underlying specific locations of individual subpopulations remain to be examined. Some known $\beta_2$AR binding partners, such as G proteins, arrestins, A-kinase anchoring proteins (AKAPs), and caveolins, co-localize with one or both subpopulations[34–37]; in particular, AKAP79 is involved in agonist-induced PKA-phosphorylation and is essential for $\beta_2$AR-induced activation of LTCC. It is also expected that both lipids and proteins that are in association with the receptor can play a role. On the basis of their subcellular distribution, PKA-phosphorylated $\beta_2$ARs may be more relevant to synapse transmission. Indeed, deletion of agonist-induced PKA-mediated phosphorylation of $\beta_2$AR completely abolishes the receptor-induced activation of LTCC in hippocampal neurons, supporting the critical role of PKA-phosphorylated $\beta_2$AR subpopulation in synaptic regulation. In contrary, due to the important role of GRKs for endocytosis of $\beta_2$AR, the GRK-phosphorylated subpopulation may be more likely involved in delivery of signal to the nucleus for gene expression[38,39].

Using single molecular approach, we gain further insight on $\beta_2$AR subpopulations phosphorylated by GRKs and PKA. Our data show that GRKs primarily phosphorylate monomeric $\beta_2$ARs whereas PKA mainly targets dimeric receptors. These data show biochemical evidence that GPCR composition may dictate post-translational modifications of individual receptors after agonist stimulation. It reveals that the dimeric $\beta_2$AR subpopulations are not only specifically modified by PKA, but also retained at the cell surface after agonist stimulation. This together with the necessary role of PKA-phosphorylated $\beta_2$ARs in activation of LTCC strongly argues the presence of functional dimeric $\beta_2$AR subpopulations in neurons. It should be noted that the single molecular analysis in this study were done using cells with over-expressing $\beta_2$AR. Although our data indicated that the composition of $\beta_2$AR was not affected by the expression levels, we could not exclude the possibility that over-expression conditions can change the stoichiometry of the receptor. Endogenously targeting of $\beta_2$AR with split fluorescent protein would be able to address this question[40].

**Fig. 4** Subpopulations of $\beta_2$AR display distinct structural properties. **a** Schematic of SiMPull with phospho-specific antibody against FLAG-mYFP-$\beta_2$AR and examples of fluorescent time traces depicting one- or two-step photobleaching of mYFP-$\beta_2$ARs, which provides stoichiometric information of $\beta_2$AR. The composition of $\beta_2$AR was calculated based on photobleaching steps in different conditions described in **b** and **c**. **b** Cells are stimulated with different concentrations of ISO for 5 min, the total $\beta_2$ARs were pulled down for photobleaching analysis, displaying a composition of ~40% dimers and ~60% monomers independent of agonist stimulation (mean ± s.d.; $n = 10, 8, 10$, and 8 independent experiments, respectively). **c** The PKA- and GRK-phosphorylated $\beta_2$ARs were pulled down with phospho-specific antibodies against S261/262 (PKA-p$\beta_2$AR) and S355/356 (GRK-p$\beta_2$AR) with different concentrations of ISO as indicated. At 1 nM, ISO only induces PKA-phosphorylation of $\beta_2$AR but not GRK-phosphorylation of $\beta_2$AR; higher concentrations of ISO induce both PKA- and GRK-phosphorylation of $\beta_2$ARs. PKA-phosphorylated $\beta_2$ARs display a composition of nearly 70% dimers and 30% monomers upon stimulation with all tested concentration of ISO (mean ± s.d.; $n = 6, 6$, and 14 independent experiments, respectively), whereas GRK-phosphorylated $\beta_2$ARs display a composition of 12% dimers and 88% monomers at high concentrations of ISO stimulation (mean ± s.d.; $n = 12$ and 21 independent experiments, respectively). N.A. not available

There are many questions remaining to be addressed. For example, while our data show that ISO does not induce dynamic changes in ratio of $\beta_2AR$ monomer and dimers, others have reported that some ligands such as ICI-118551 and Carvedilol are capable to induce dynamic changes in monomer and dimer ratio of the receptor[41,42]. In the same vein, we speculate that certain ligands may be able to induce dynamic changes in GRK and PKA-phosphorylated populations. Other interesting questions include why monomeric receptors are more sensitive to GRKs than PKA and vice versa and whether phosphorylation leads to change in ratio of $\beta_2AR$ monomer and dimers. Potential

explanations may include: the pre-coupling of different kinase (and scaffold proteins) with receptors; different accessibility of two kinases (receptor intracellular loops vs. C-terminal); and different locations at lipid rafts or non-rafts microdomains. Moreover, it remains to be determined whether neurons maturation affects the monomer and dimer ratio of the receptor. These are the questions that we look forward to pursuing in the near future.

Together, these data provide molecular mechanisms on the existence and functional relevance of distinct subpopulations of $\beta_2AR$. It is conceivable that PKA- and GRK-phosphorylated

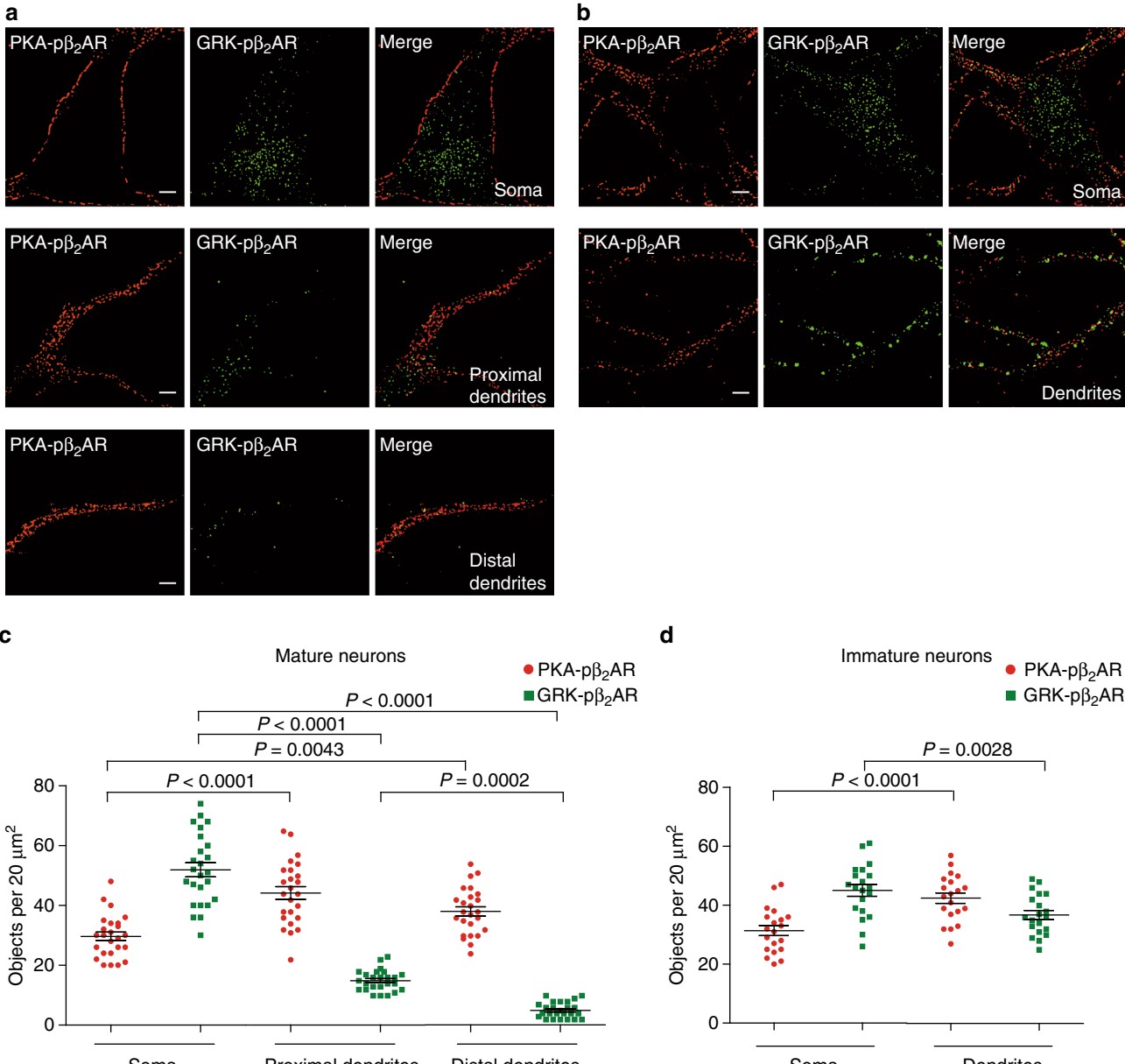

**Fig. 5** Subpopulations of $\beta_2AR$ are spatially segregated between dendrites and soma in hippocampal neurons. Primary hippocampal neurons expressing FLAG-$\beta_2AR$ at 18–21 days in vitro (DIV, mature, **a**) or 6–8 DIV (immature, **b**) were treated with 1 μM ISO for 5 min (**a**) or 10 min (**b**). Neurons were stained with phospho-specific antibodies against S261/262 (PKA-p$\beta_2AR$) and S355/356 (GRK-p$\beta_2AR$) as indicated to show phosphorylated $\beta_2ARs$ in soma, and proximal and distal dendrites (Scale bar, 2 μm. $n = 25$ and 20 cells, respectively, three independent experiments). Representative SIM images in **a** and **b** were from one single neuron. **c** Numbers of fluorescent objects per 20 μm$^2$ regions were quantified with ImageJ in 18–21 DIV mature neurons (PKA-p$\beta_2AR$, soma 29.6 ± 1.7, proximal dendrites 42.6 ± 2.2, and distal dendrites 38.6 ± 1.8; GRK-p$\beta_2AR$, soma 50.4 ± 2.7, proximal dendrites 14.9 ± 0.7, and distal dendrites 5.1 ± 0.6). **d** Numbers of fluorescent objects per 20 μm$^2$ regions were quantified with ImageJ in 6–8 DIV immature neurons (PKA-p$\beta_2AR$, soma 31.4 ± 1.7 and dendrites 42.6 ± 1.8; GRK-p$\beta_2AR$, soma 45.1 ± 2.0, and dendrites 36.8 ± 1.5). Error bars denote s.e.m., multiplicity adjusted $P$ values are computed by one-way ANOVA followed by Tukey's test between indicated groups

subpopulations can lead to divergent downstream intracellular signals in space and time, offering a molecular basis on biased signaling transduced from activation of $\beta_2AR$ in a single cell. This study unveils a paradigm that can be extended to study other GPCRs in general. It also offers a structural platform to discover and study biased drugs that selectively modulate individual subpopulations of a GPCR.

## Methods

**Cell culture and transfection.** Human embryonic kidney HEK293 cells were from American Type Culture Collection (ATCC) and were maintained in Dulbecco's modified Eagle medium (Mediatech, VA) supplemented with 10% fetal bovine serum (Sigma, MO). Primary mouse hippocampal neurons were isolated and cultured from early postnatal (P0-P1) wild type, $\beta_1AR$ knockout (KO), $\beta_2AR$ KO and $\beta_1AR/\beta_2AR$ double knockout (DKO) mouse pups, and primary rat hippocampal neuronal cultures were prepared from E17-E19 embryonic rats[43,44]. Briefly, dissected hippocampi were dissociated by 0.25% trypsin treatment and trituration. Neurons were plated on poly-D-lysine-coated (Sigma, MO) glass coverslips for imaging and 6-well plate for biochemistry at a density of 7500 cells/cm² and 10,000 cells/cm², respectively. Neurons were cultured in Neurobasal medium supplemented with GlutaMax and B-27 (Thermo Scientific, MA).

Animal protocols were approved by IACUC of the University of California at Davis according to NIH regulations.

HEK293 cells were transfected with plasmids using polyethylenimine (PEI) according to manufacturer's instructions (Sigma, MO). Neurons were transfected by the $Ca^{2+}$-phosphate method[45]. Briefly, cultured neurons at either 6–8 days in vitro (DIV) or 18–21 DIV were switched to pre-warmed Eagle's minimum essential medium (EMEM, Thermo Scientific, MA) supplemented with GlutaMax 1 h before transfection, conditioned media were saved. DNA precipitates were prepared by 2×HBS (pH 6.96) and 2 M $CaCl_2$. After incubation with DNA precipitates for 1 h, neurons were incubated in 10% $CO_2$ pre-equilibrium EMEM for 20 min, then replaced with conditioned medium.

**Antibodies and chemicals.** Mouse monoclonal antibodies against $\beta_2AR$ at serine 261/262 (clone 2G3 and 2E1) and at serine 355/356 (clone 10A5) were kindly provided by Dr. Richard Clark (UT Huston). Polyclonal antibodies against $\beta_2AR$ (m20 and sc-570) and phosphorylated $\beta_2AR$ at serine 355/356 (sc-22191R and sc-16719R) were purchased from Santa Cruz Biotechnology (SCBT, CA). Polyclonal antibodies against $\alpha_1 1.2$ residues 754–901 for total $\alpha_1 1.2$ (FP1), $\alpha_1 1.2$ residues 1923–1935 for phosphorylated serine 1928 site (LGRRApSFHLECLK, pS1928) and $\alpha_1 1.2$ residues 1694–1709 for phosphorylated serine 1700 site (EIRRAIPsGDL-TAEEEL, pS1700) were made in house[26]. Other antibodies used in the experiments include: anti-FLAG-M1 and biotinylated anti-FLAG-M2 (F3040 and F9291, Sigma, MO), biotinylated goat anti-mouse IgG, and goat anti-rabbit IgG (111-065-144 and 115-055-166, Jackson ImmunoResearch, PA).

Isoproterenol (ISO) (Sigma, MO) was freshly prepared in water for every experiment. When necessary, ICI-118551 (Sigma, MO) was prepared in water as a stock solution and added into solution to a final concentration of 1 µM.

**Plasmids and viruses.** Plasmids containing FLAG-tagged mouse $\beta_2AR$ (FLAG-$\beta_2AR$) and its phosphorylation-deficient mutants lacking pS261/262 or pS355/356 (FLAG-$\beta_2AR$-PKAmut or FLAG-$\beta_2AR$-GRKmut) were described before[46]. DNA

constructs containing HA-tagged rat L-type calcium channel (LTCC) $\alpha_1 1.2$, $\beta_2 a$ and $\alpha_2 \delta$ subunits were described elsewhere[26]. Plasmids containing monomeric CFP-CD86 and constitutively dimeric CFP-CD28 were kindly provided by Dr. Moritz Bunemann (Philipps University of Marburg, Germany), in which CFP was replaced by FLAG-mYFP.

FLAG and FLAG-mYFP-tagged $\beta_2AR$ and its phosphorylation-deficient mutants were inserted into lentiviral vector pLenti-H1-CAG (kindly provided by Dr. Sergi Simo, UC Davis), and were used to produce neuron-preferential lentivirus according to a method described elsewhere[47]. Neurons at 10 DIV were infected by same titer of lentiviruses as indicated, and were used in single-molecule pulldown analysis at 14 DIV.

**SIM and confocal microscope imaging.** HEK293 cells growing on poly-D-lysine-coated coverslips were transfected with FLAG-$\beta_2AR$ or one of its mutants. Rat hippocampal neurons growing on poly-D-lysine-coated coverslips were transfected with FLAG-$\beta_2AR$ at either 6–8 DIV or 18–21 DIV.

Cells were serum-starved for 2 h and stimulated with 1 µM ISO at indicated times. Cells were then fixed, permeabilized, and co-stained with indicated antibodies with a final concentration of 1 µg/ml for each antibody, which were revealed with a 1:1000 dilution of Alexa fluor 488 or Alexa fluor 594 conjugated goat anti-mouse or anti-rabbit IgG antibodies, respectively (A-11032, A-11037, A-11029, and A-11034, Life technologies, CA). Fluorescence images were taken by Zeiss LSM 700 confocal microscope with a ×63/1.4 numerical aperture oil-immersion objective lens or Nikon 3D structured illumination (N-SIM) super-resolution microscope with a ×100/1.49 numerical aperture TIRF oil-immersion objective lens (MCB Imaging Facility, UC Davis). For SIM data, each sample was imaged from three angles and each angle was phase-shifted 5 times, 15 raw data images were then computed to a reconstructed super-resolution image. Quantitative image analysis was carried out on unprocessed images using ImageJ software (http://rsb.info.nih.gov/ij). Co-localization analysis was assessed by calculating the Pearson's correlation coefficient between two-color channels in the indicated images using the co-localization plug-in for ImageJ. Numbers of fluorescent objects within each 20 µm² regions in images of neurons were quantified using the Squassh plug-in for ImageJ[48].

**SiMPull imaging and quantification.** HEK293 cells transfected with mYFP-$\beta_2AR$ alone, or together with LTCC $\alpha_1 1.2$, $\beta_2 a$, and $\alpha_2 \delta$, were either untreated or treated with different concentrations of ISO for 5 min, then harvested into lysis buffer (10 mM Tris pH 7.9, 0.1% DDM, 150 mM NaCl, 2 mM EDTA) with protease and phosphatase inhibitor cocktail (final concentration 10 µM pepstatin, 1 mM PMSF, 1 mM NaF, 1 mM $Na_3VO_4$, 25 mM $Na_2O_7P_2$, and 10 mM β-glycerophosphate). Clarified lysates were diluted and used for single-molecule pulldown (SiMPull) analysis. Proteins were immobilized on SiMPull slides by 10 nM of biotinylated anti-FLAG, goat anti-mouse IgG or goat anti-rabbit IgG together with 5 nM phospho-specific $\beta_2AR$ antibodies as indicated. Immobilized proteins were visualized by a prism-type total internal reflection fluorescence (TIRF) microscope.

For the $\beta_2AR$ and $\alpha_1 1.2$ dissociation test, DKO mouse hippocampal neurons were infected with lentirviruses that express WT or mutant FLAG-mYFP-$\beta_2AR$ for 4 days then harvested using the same procedure as for HEK293 cells. Lysates were tested in SiMPull using 10 nM biotinylated goat anti-rabbit IgG together with 5 nM FP1 antibody to pulldown endogenous $\alpha_1 1.2$ and its associated $\beta_2AR$.

Mean spot count per image and standard deviation were calculated from images taken from 10–20 different regions using scripts written in Matlab.

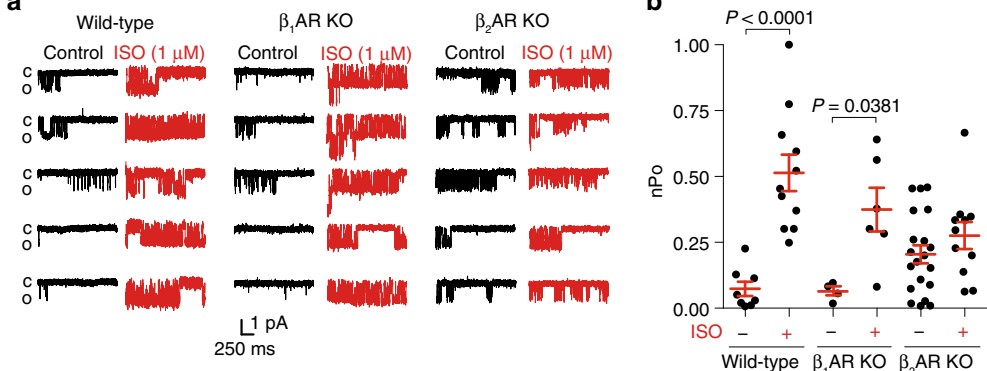

**Fig. 6** $\beta_2AR$ is necessary for ISO-induced upregulation of LTCC $Ca_v 1.2$ activity in hippocampal neurons. **a** Representative single-channel recordings of LTCC $Ca_v 1.2$ currents in hippocampal neurons from wild type (WT), $\beta_1AR$ knockout (KO) and $\beta_2AR$ KO mice at 7–14 DIV after depolarization from −80 to 0 mV without (black traces) and with stimulation of 1 µM ISO (red traces) in the patch pipette. **b** Data in **a** were quantified and plotted. The ISO-induced increases in LTCC overall channel activity (nPo) were detected in WT and $\beta_1AR$ KO neurons, but absent in $\beta_2AR$ KO neurons ($n = 8, 11, 4, 6, 20$, and 11 cells, respectively). Error bars denote s.e.m., exact $P$ values are computed by Mann–Whitney test

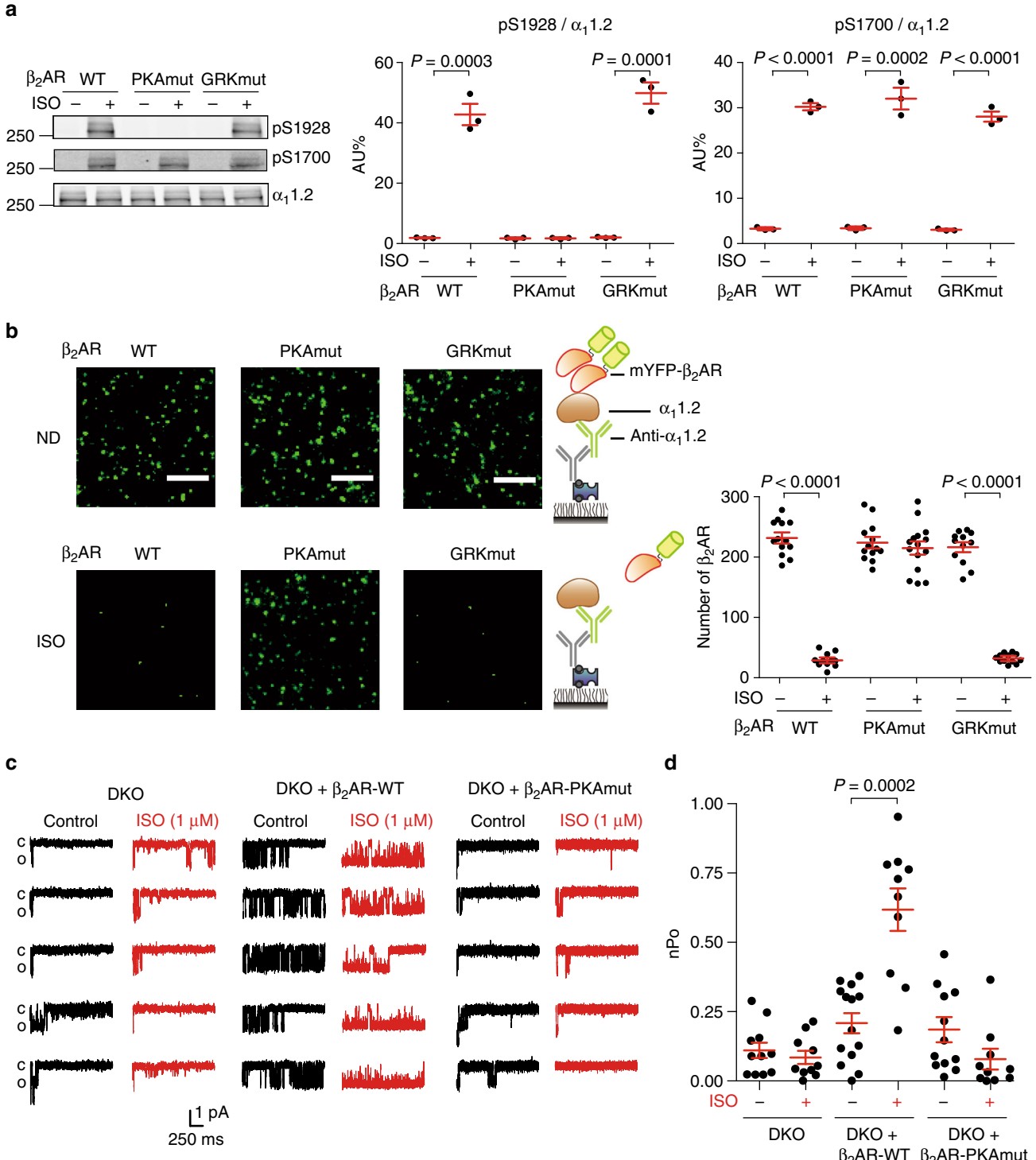

**Fig. 7** PKA-pβ2AR is necessary for activation of β2AR-α11.2 complex in hippocampal neurons. WT or mutant β2AR lacking either PKA sites (PKAmut) or GRK sites (GRKmut) was expressed in hippocampal neurons lacking both β1AR and β2AR genes (DKO). **a** Neurons were either not stimulated (ND) or stimulated with 1 μM ISO for 5 min. The total α11.2 was immunoprecipitated; the phosphorylation of α11.2 at S1928 and S1700 was probed with phospho-specific antibodies and normalized to total α11.2. Representative of three independent experiments. Molecular weight markers (in kDa) are indicated on the left. **b** The endogenous α11.2 was pulled down in SiMPull as depicted; representative images show that mYFP-β2AR pulled down together with α11.2. Scale bar, 5 μm. Quantification of the numbers of mYFP-β2AR bound to α11.2 shows that PKA-phosphorylation of β2AR is required for dissociation of the β2AR-α11.2 complex. Representative of $n = 13/12$, 13/15, and 12/12 images for WT, PKAmut, and GRKmut β2AR groups, four independent experiments. **c** DKO hippocampal neurons at 7–14 DIV expressing WT and mutant β2AR were subjected to single-channel recording of LTCC currents using the same method as Fig. 6. **d** The overall channel activity (nPo) of LTCC Ca$_v$1.2 was quantified from **c** ($n = 11$, 10, 14, 10, 12, and 10 cells, respectively). Error bars denote s.e. m., exact $P$ values are computed by Student's $t$-test in **a** and **b**, and by Mann–Whitney test in **d**

**SiMPull co-localization and photobleaching analysis**. Single-molecule co-localization between mYFP and mCherry was performed using scripts written in Matlab[18,49]. Briefly, two separate images of the same region were taken using mYFP and mCherry excitation. The fluorescent spots in both images were fit with Gaussian profiles to determine the center positions of mYFP and mCherry molecules to half-pixel accuracy. The mCherry and mYFP molecules within a 1-pixel distance (~150 nm) were considered as co-localized. The overlap percentage was calculated as the number of co-localized mYFP molecules divided by the total number of mYFP molecules.

Single-molecule fluorescence time traces of immobilized mYFP-tagged proteins were manually scored for the number of bleaching steps by a well-established method[18–21]. To avoid false co-localization, samples were immobilized at an optimal surface density (100–400 molecules in a 2000 mm² imaging area). The number of photobleaching steps (single frame intensity drops of equal size) in each trace was manually determined. The fluorescence trace of each molecule was classified as having 1–4 bleaching steps or was discarded if no clean bleaching steps could be identified (Supplementary Fig. 3). At least 1000 molecules were analyzed for each condition. The population distribution of observed bleaching events and discarded traces is reported in Supplementary Table 1.

**Determination of stoichiometry of fluorescent proteins**. mYFP molecules are about 75% fluorescent active (visible) and 25% fluorescent inactive (invisible) in cells[20,21], this means that a constitutively dimeric mYFP-CD28 will statistically yield 56.25% 2-steps photobleaching events when both mYFP are active in the dimers; and 37.5% 1-step photobleaching events when only one copy of mYFP are fluorescently active in the dimers; and 6.25% dark events when both mYFP are fluorescently inactive in the dimers, which are undetectable in SiMPull. Accordingly, the percentage of monomer and dimer of a target protein was calculated by fitting bleaching data with the equation:

$$P_{dimer} = \frac{7500 \times Q_{2-steps}}{56.25 - 18.75 \times Q_{2-steps}}, P_{monomer} = 100 - P_{dimer},$$

$P_{dimer}$ is the percentage of dimer of a target protein, $P_{monomer}$ is the percentage of monomer of a target protein, and $Q_{2-steps}$ is the quotient of the numbers of 2-steps bleaching events divided by numbers of overall bleaching events.

**Immuno-isolation and co-immunoprecipitation**. For immuno-isolation tests, HEK293 cells stably expressing FLAG-$\beta_2$AR were serum-starved for 2 h and stimulated with 1 μM ISO for 30 s or 10 min, then harvested by lysis buffer (10 mM Tris pH 7.4, 1% NP40, 150 mM NaCl, 2 mM EDTA) with protease and phosphatase inhibitor cocktail. Clarified lysates were incubated with anti-pS355/356 $\beta_2$AR (sc-16719R) and protein-A agarose (Thermo Scientific, MA) overnight at 4 °C. The beads were collected as the first immuno-isolation. Subsequently, the supernatants were further incubated with anti-FLAG-M2 agarose (Sigma, MO) for 2 h at 4 °C. These beads were collected as the second immuno-isolation. A control of total FLAG-$\beta_2$AR was collected by directly incubating clarified lysates with anti-FLAG-M2 agarose.

For $\beta_2$AR and $\alpha_1$1.2 dissociation test in HEK293 cells, FLAG-$\beta_2$AR was co-transfected with $\alpha_1$1.2, $\beta_2$a, and $\alpha_2\delta$ subunits using a standard calcium phosphate method. Cells were serum-starved overnight supplemented with 1 μM ICI-118551 and 10 μM isradipine. On the next day cells were washed with warmed serum-free medium and treated with 10 μM ISO for 10 min, and then harvested by lysis buffer with protease and phosphatase inhibitor cocktail. Clarified lysates were incubated with anti-FLAG-M2 agarose for 2 h at 4 °C. Proteins were eluted from beads with 2 × SDS non-reducing buffer.

Protein samples were analyzed by Western blot using antibodies as indicated at a 1:1000 dilution and signals were detected by Odyssey scanner (Li-cor, NE). Uncropped versions of the scans are presented in Supplementary Figs. 7 and 8b.

**Plasma membrane and endosomal fractionation**. HEK293 cells over-expressing FLAG-$\beta_2$AR were surface labeled by sulfo-NHS-LC-LC-biotin at 4 °C according to manufacturer's instructions (Thermo Scientific, MA) followed by 1 nM or 1 μM ISO stimulation for 10 min in warmed medium. Samples were then immediately placed into an ice bath, washed with ice-cold PBS and lysed by hypotonic buffer (10 mM Tris pH 7.4, 10 mM KCl, 1.5 mM MgCl₂, 2 mM EDTA, with protease and phosphatase inhibitor cocktail). After removal of debris and nuclei by centrifugation at 1000 g for 10 min, streptavidin beads (Invitrogen, CA) were added into lysates to pulldown biotin-labeled PM proteins for 2 h at 4 °C. The intracellular organelles in the supernatant were then lysed to expose endocytosed biotin-labeled FLAG-$\beta_2$AR by addition of detergent NP40 (1%) and NaCl (150 mM). Streptavidin beads were added and incubated for another 2 h at 4 °C to collect the endosomal fraction. Beads were washed and eluted by 2×SDS loading buffer. The biotinylated proteins were analyzed by Western blot using antibodies at a 1:1000 dilution as indicated. Uncropped versions of the scans are presented in Supplementary Fig. 8a.

**Cell-attached patch clamp electrophysiology**. Primary mouse hippocampal neurons were used at 7–14 DIV. Cell-attached patch clamp recordings were performed on an Olympus IX70 inverted microscope in a 15-mm culture coverslip at room temperature (22–25 °C). Signals were recorded at 10 kHz and low-pass filtered at 2 kHz with an Axopatch 200B amplifier and digitized with a Digidata 1440 (Molecular Devices). Recording pipettes were pulled from borosilicate capillary glass (0.86 OD) with a Flaming micropipette puller (Model P-97, Sutter Instruments) and polished (polisher from World Precision Instruments). Pipette resistances were strictly maintained between 6–7 MΩ to ameliorate variations in number of channels in the patch pipette. The patch transmembrane potential was zeroed by perfusing cells with a high K⁺ extracellular solution containing (in mM) 145 KCl, 10 NaCl, and 10 HEPES, pH 7.4 (NaOH). The pipette solution contained (in mM) 20 tetraethylammonium chloride (TEA-Cl), 110 BaCl₂ (as charge carrier), and 10 HEPES, pH 7.3 (TEA-OH). This pipette solution was supplemented with 1 μM ω-conotoxin GVIA and 1 μM ω-conotoxin MCVIIC to block N and P/Q-type Ca²⁺ channels, respectively, and (S)-(−)-BayK-8644 (500 nM) was included in the pipette solution to promote longer open times and resolve channel openings. Indeed, BayK-8644 is routinely used to augment detection of L-type channels in single-channel recordings[26,27,50]. To examine the effects of β-adrenergic stimulation on the L-type Ca$_V$1.2 single-channel activity, 1 μM isoproterenol was added to the pipette solution in independent experiments. Note that we have previously used the L-type Ca$_V$1.2 channel blocker nifedipine (1 μM) to confirm the recording of L-type Ca$_V$1.2 currents under control conditions and in the presence of isoproterenol[26]. Single-channel activity was recorded during a single pulse protocol (2 s) from a holding potential of −80 mV to 0 mV every 5 s. An average of 50 sweeps were collected with each recording file under all experimental conditions. The half-amplitude event-detection algorithm of pClamp 10 was used to measure overall single-channel L-type Ca$_V$1.2 activity as nPo, where n is the number of channels in the patch and Po is the open probability. Note that the number of channels in each patch recording (n) was not estimated and that all data are presented as "nPo" (product of n and channel open probability). nPo values were pooled for each condition and analyzed with GraphPad Prism software.

**Statistical analysis**. Data were analyzed using GraphPad Prism software and expressed as mean ± s.d. or mean ± s.e.m. as indicated in figure legends. Differences between two groups were assessed by appropriate two-tailed unpaired Student's t-test or nonparametric Mann–Whitney test. Differences among three or more groups were assessed by one-way ANOVA with Tukey's post hoc test. $P < 0.05$ was considered statistically significant.

**Data availability**. The data that support the findings of this study are available from the corresponding author upon reasonable request.

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

## Acknowledgements

This study was supported by NIH grants HL127764 and HL112413 and VA Merit grant 01BX002900 to Y.K.X., HL098200 and HL121059 to M.F.N., NS078792 and AG055357 to J.W.H. A.S., Q.S., and D.C. are recipients of American Heart Association postdoctoral fellowship. Y.K.X. is an established American Heart Association investigator.

## Author contributions

A.S. and Y.K.X. conceived and designed experiments. A.S., M.F.N., M.N.-C. and Q.S. performed experiments. Y.D. helped in data analysis. D.C. provided hippocampal neuron culture. J.Q. provided analytic tools. A.S. and Y.K.X. analyzed data and wrote the manuscript with inputs from M.F.N. and J.W.H. Y.K.X. provided overall project supervision.

## Additional information

**Competing interests:** The authors declare no competing interests.

