## [Peer Review File · Nature Communications]

Reviewers' comments:

Reviewer #1 (Remarks to the Author):

Review on Shen Nature Comm

In the submitted work Shen et al. employs SIM imaging and imaging based single molecules pull down analysis to investigate the role of agonist induced phosphorylation in the spatial organization of the beta adrenergic receptor. Their work indicates that upon agonist induction receptors that are phosphorylated by PKA are spatially distinct from receptors that are phosphorylated by GRK. Moreover, while PKA phosphorylated receptors are predominantly monomeric GRK phosphorylated receptors mostly exist in dimers. The distribution of PKA- and GRK- phosphorylated receptors was also different in hippocampal neurons and PKA phosphorylation was shown to be necessary for dissociation and activation of the β 2AR- α 11.2 complex. Overall, the experiments are suited for the biological question, well performed and support the conclusions. I only have minor comments.

Specific comments:

1. All β 2AR labeling experiments were done on cells treated with ISO. It will be good to show β 2AR distribution in the absence of ISO as well.
2. In SIM images – it should be clearly indicated in the figure legend how many cells were imaged and how many times was the experiment repeated. Also, what is the biological variability? Do 100% of the cells exhibit this phenotype? Part of this information may be represented in the numbers written inside the bars of some of the plots associated with the images. However, I could not find anywhere in the text what these numbers represents.
3. A more detailed explanation on the photobleaching experiments that are used for determining the subunits stoichiometry should be provided in main text. I realize that this information is included in the online method section and is based on a previously published method but because this is not a trivial or a widely used approach I think it should be explained to the readers in the main text.
4. The cartoons presented in figure 3b and 5b are somewhat misleading because they include both a schematic description of the assay and the conclusion drawn from the experiment. This should be clearly indicated in the text. Also, to avoid this confusion it will be better to place the cartoons after the fluorescence images and not before.
5. Figure 3c should be placed after 3b and not before.
6. In theory, applying STORM microscopy would be more suitable to this project than SIM. However, since the phenotypes are clearly observed by SIM, increased resolution is not needed to answer the questions addressed in this work.

Reviewer #2 (Remarks to the Author):

This is an interesting and novel report showing that the phosphorylation state of the β 2-adrenergic receptor is correlated with its ability to have distinct, spatially segregated functions in a single cell. The data are strong, the results seem well justified, and are likely to stimulate future investigations. The molecular mechanisms by which signaling molecules such as GPCRs, can have distinct roles at the single cell level need to be studied in more detail. this is an important problem - what are the mechanisms that maintain segregated signaling pathways in individuals cells that use the same receptor. There are many factors that influence the formation of signaling complexes but this report adds to the literature and will have the field thinking more about differential phosphorylation and agonist-dependent subcellular localization and/or differential rates of internalization from the plasma membrane.

Comments

Introduction and Discussion

- The authors unnecessarily overstate their case somewhat in the introduction. While much remains to be discovered, these sentences seem too extreme "how a single ligand-receptor pair can selectively transduce different signaling pathways in time and space in a single cell is unclear" and "the molecular basis for biased ligands that selectively promote specific intracellular signaling remain unclear" (p. 2). e.g. β -arrestin function dependent on the phosphorylation pattern or 'barcode' of the receptor: <https://doi.org/10.1016/j.ceb.2013.10.008>, and other examples: DOI: 10.1126/science.1232808, <https://doi.org/10.1016/j.it.2014.02.004>, <https://doi.org/10.1016/j.ceb.2013.10.008>

Results

- Iso-induced segregation of β_2 -ARs at the plasma membrane. The authors should include data (at least in Fig. 1) to show pre-agonist distribution so that the reader has a point of comparison.
- Fig. 1. P. 3. What does "readily co-localize" mean in the context of data shown in Fig. 1a? The use of readily implies that co-localization is fast or occurs easily relative to something else?
- Fig. 2. The authors title for fig 2 is "distinct membrane trafficking after prolonging stimulation". Elsewhere the authors refer to more rapid internalization of GRK- β_2 -AR from the membrane. I'd suggest changing the title to be more consistent with the data.
- Fig. 4. P. 5. The description of figure 4 is confusing. PKA- β_2 -AR are found in soma and dendrites of immature and mature neurons. GRK- β_2 -AR are in much lower abundance in dendrites of mature neurons as compared to soma and also to PKA- β_2 -AR. The authors refer to lack of co-localization "...images showing lack of co-localization between PKA- and GRK-phospho β_2 AR". It's difficult to determine if there is less co-localization from the images shown in Fig. 4 and so referring to co-localization in this context is confusing. The term localization was used in earlier figures as a measure of overlap of the signals when both are present. If the authors are relying on supplemental data to comment on co-localization then they should include these data. Otherwise, I suggest changing wording for clarity. E.g. both PKA- β_2 -AR and GRK- β_2 -AR are found in soma of mature neurons but in dendrites, GRK- β_2 -AR levels are much lower compared to PKA- β_2 -AR. i.e. that they are in different subcellular compartments in mature neurons.
- Fig. 5. The authors should include more information about how they performed the single channel experiments and analyses. Was Isoproterenol added to the same patch recording or are the authors comparing across different patches? If the latter, how do they control for differences in channel numbers across patches when there is no internal control? With Bay K present it should be possible to measure the total number of channels in the patch by stepping to a very positive value from a negative holding potential. Was this done?
- I am not a fan of supplementary data. I realize this is an editorial decision but, if data are essential to support the conclusions of the report then I would argue they should be include in the manuscript. I recommend that the authors review their 8 figures of supplementary information - this seems excessive.
- I recommend that the authors convert all histograms to plots that include individual data points. This is so much more informative and - as shown nicely in Fig. 4 - the reader can access uncompressed data.
- Depending on this journal's policy - I recommend removing *, **, *** symbols and rather include absolute p values in the figure legend.
- "opening probability" = frequency of opening "open probability" = the probability of the channel being open. I believe that the latter is what you are measuring.
- "simulation" to "stimulation"

Reviewer #3 (Remarks to the Author):

This manuscript ties together observations made over the years by the Yang and Hell labs. It is well written, with well-controlled experiments using state-of-the-art methodology. For the first time I think, we can now see the difference between beta2AR populations phosphorylated by PKA or by GRKs- even though we have known for years that both protein kinases call receptor desensitization. The authors show that GRK mainly phosphorylates a pool of monomeric receptors that internalizes and that PKA phosphorylates a receptor dimer which remains at the plasma membrane as part of a larger signalling complex that contains effector molecules like the L-type calcium channel.

They show that these monomeric and dimeric receptor complexes are spatially segregated using super resolution microscopy combined with single molecule pulldowns. They had already shown that about 50% of the beta2AR was dimeric using this latter technique and validate those results here both in terms of simple spectral overlap and with step photobleaching approaches.

Moving beyond HEK 293 cells, they show distinct distributions in hippocampal neurons with GRK-sensitive receptors in the soma and PKA-sensitive receptors in dendrites- which occurred during neuronal maturation.

Finally, they show that the PKA-sensitive receptors (in neurons with a WT beta1 and beta2AR KO background) are necessary for modulation of L-type calcium channels.

The data are generally convincing and I enjoyed reading this manuscript. My only criticism as such is it would be interesting to have more discussion about how these receptors adopt their fate in the first place. Why are monomeric receptors more sensitive to GRK than PKA and vice versa? Are the populations capable of dynamic exchange? What is it about neuronal maturation that alters this? I am not asking for additional experiments although the authors could be encouraged to manipulate anterograde receptor trafficking with their mutant receptors. I feel that this paper might be seminal at some point and I just want to help them get it right!

We would like to thank all Reviewers for their thorough and helpful feedbacks to improve the manuscript. A point by point response to Reviewers' comments is given below.

Reviewer #1 (Remarks to the Author):

In the submitted work Shen et al. employs SIM imaging and imaging based single molecules pull down analysis to investigate the role of agonist induced phosphorylation in the spatial organization of the beta-adrenergic receptor. Their work indicates that upon agonist induction receptors that are phosphorylated by PKA are spatially distinct from receptors that are phosphorylated by GRK. Moreover, while PKA phosphorylated receptors are predominantly monomeric GRK phosphorylated receptors mostly exist in dimers. The distribution of PKA- and GRK- phosphorylated receptors was also different in hippocampal neurons and PKA phosphorylation was shown to be necessary for dissociation and activation of the β 2AR- α 11.2 complex. Overall, the experiments are suited for the biological question, well performed and support the conclusions. I only have minor comments.

Specific comments:

1. All β 2AR labeling experiments were done on cells treated with ISO. It will be good to show β 2AR distribution in the absence of ISO as well.

Response: We thank the reviewer for the suggestion. The data to show β 2AR distribution in the absence of ISO has been added to the new Figure 1a.

2. In SIM images – it should be clearly indicated in the figure legend how many cells were imaged and how many times was the experiment repeated. Also, what is the biological variability? Do 100% of the cells exhibit this phenotype? Part of this information may be represented in the numbers written inside the bars of some of the plots associated with the images. However, I could not find anywhere in the text what these numbers represents.

Response: We have now used scatter dot plots to show the number of cells measured in SIM images, and we also have included cell numbers and experimental repeats in the figure legends. In these experiments, we observed that all of the cells exhibit this phenotype ---- after ISO treated for 5-10 minutes, in a total of 31 cells from 3 repeats we have measured, no cell displays significant intracellular staining of PKA-phosphorylated β 2AR that has a comparable level to that of intracellular staining of GRK-phosphorylated β 2AR. This comparison has been added to the text and the new Figure 2b.

3. A more detailed explanation on the photobleaching experiments that are used for determining the subunits stoichiometry should be provided in main text. I realize that this information is included in the online method section and is based on a previously

published method but because this is not a trivial or a widely used approach I think it should be explained to the readers in the main text.

Response: We have now included more detail description of photobleaching experiments in the main text (page 5).

4. The cartoons presented in figure 3b and 5b are somewhat misleading because they include both a schematic description of the assay and the conclusion drawn from the experiment. This should be clearly indicated in the text. Also, to avoid this confusion it will be better to place the cartoons after the fluorescence images and not before.

Response: We have now moved the cartoons after the data to avoid confusion.

5. Figure 3c should be placed after 3b and not before.

Response: The panels have been rearranged as suggested.

6. In theory, applying STORM microscopy would be more suitable to this project than SIM. However, since the phenotypes are clearly observed by SIM, increased resolution is not needed to answer the questions addressed in this work.

Response: We thank the reviewer for insightful comments. Initially we have tried STORM microscopy with Dr. Bo Huang in UCSF, a leader in STORM imaging field. However, we ran into high background issues in dual color labeling. We are glad that the SIM image is sufficient to differentiate two colors in our experiments.

Reviewer #2 (Remarks to the Author):

This is an interesting and novel report showing that the phosphorylation state of the b2-adrenergic receptor is correlated with its ability to have distinct, spatially segregated functions in a single cell. The data are strong, the results seem well justified, and are likely to stimulate future investigations. The molecular mechanisms by which signaling molecules such as GPCRs, can have distinct roles at the single cell level need to be studied in more detail. this is an important problem - what are the mechanisms that maintain segregated signaling pathways in individuals cells that use the same receptor. There are many factors that influence the formation of signaling complexes but this report adds to the literature and will have the field thinking more about differential phosphorylation and agonist-dependent subcellular localization and/or differential rates of internalization from the plasma membrane.

Comments:

Introduction and Discussion

- The authors unnecessarily overstate their case somewhat in the introduction. While much remains to be discovered, these sentences seem too extreme "how a single

ligand-receptor pair can selectively transduce different signaling pathways in time and space in a single cell is unclear” and “the molecular basis for biased ligands that selectively promote specific intracellular signaling remain unclear” (p. 2). e.g. β -arrestin function dependent on the phosphorylation pattern or ‘barcode’ of the receptor: <https://doi.org/10.1016/j.ceb.2013.10.008>, and other examples:

DOI: 10.1126/science.1232808, <https://doi.org/10.1016/j.it.2014.02.004>,
<https://doi.org/10.1016/j.ceb.2013.10.008>

Response: We thank the reviewer for the critical comment. The statement has been revised accordingly and the references have been cited.

Results

- Iso-induced segregation of β 2-ARs at the plasma membrane. The authors should include data (at least in Fig. 1) to show pre-agonist distribution so that the reader has a point of comparison.

Response: We thank the reviewer for the suggestion. The data to show β 2AR distribution pre-agonist stimulation has been added to the new Figure 1a.

- Fig. 1. P.3. What does “readily co-localize” mean in the context of data shown in Fig. 1a? The use of readily implies that co-localization is fast or occurs easily relative to something else?

Response: The word “readily” has been removed.

- Fig. 2. The authors title for fig 2 is “distinct membrane trafficking after prolonging stimulation”. Elsewhere the authors refer to more rapid internalization of GRK-p β 2-AR from the membrane. I’d suggest changing the title to be more consistent with the data.

Response: The phrase “after prolonging stimulation” has been removed from the title to keep consistent with data and the other figure titles.

- Fig. 4. P. 5. The description of figure 4 is confusing. PKA-p β 2-AR are found in soma and dendrites of immature and mature neurons. GRK-p β 2-AR are in much lower abundance in dendrites of mature neurons as compared to soma and also to PKA-p β 2-AR. The authors refer to lack of co-localization “...images showing lack of co-localization between PKA- and GRK-phospho β 2AR”. It’s difficult to determine if there is less co-localization from the images shown in Fig. 4 and so referring to co-localization in this context is confusing. The term localization was used in earlier figures as a measure of overlap of the signals when both are present. If the authors are relying on supplemental data to comment on co-localization then they should include these data. Otherwise, I suggest changing wording for clarity. E.g. both PKA-p β 2-AR and GRK-p β 2-AR are found in soma of mature neurons but in dendrites,

GRK-pb2-AR levels are much lower compared to PKA-pb2-AR. i.e. that they are in different subcellular compartments in mature neurons.

Response: We thank the reviewer for the critical comments. The text has been revised to avoid the confusion between the co-localization (overlap) on the same membrane domain and the spatial segregation between soma and dendrites. Now Figure 4 data are solely used for the measure of PKA- and GRK-pb2-AR amounts in soma and dendrites. And we have added quantification of overlaps in Supplementary Figure 4c for the measure of co-localization in neurons.

- Fig. 5. The authors should include more information about how they performed the single channel experiments and analyses. Was Isoproterenol added to the same patch recording or are the authors comparing across different patches? If the latter, how do they control for differences in channel numbers across patches when there is no internal control? With Bay K present it should be possible to measure the total number of channels in the patch by stepping to a very positive value from a negative holding potential. Was this done?

Response: We thank the reviewer for this suggestion. We have expanded the cell-attached patch clamp electrophysiology method section to include additional information about these experiments.

- I am not a fan of supplementary data. I realize this is an editorial decision but, if data are essential to support the conclusions of the report then I would argue they should be included in the manuscript. I recommend that the authors review their 8 figures of supplementary information - this seems excessive.

Response: We thank the reviewer for this suggestion. We have now included the data from the old Supplementary Figure 2 into the new Figure 1c, and moved old Supplementary Figure 7 to the new Figure 5a-b.

- I recommend that the authors convert all histograms to plots that include individual data points. This is so much more informative and – as shown nicely in Fig. 4 – the reader can access uncompressed data.

Response: Now all the graphs have been converted to scatter dot plots. We also have included n numbers and experimental repeats in the figure legends.

- Depending on this journal's policy - I recommend removing *, **, *** symbols and rather include absolute p values in the figure legend.

Response: We have removed * symbols and now used exact P values in all figures.

- “opening probability” = frequency of opening “open probability” = the probability of the channel being open. I believe that the latter is what you are measuring.

Response: The words have been revised accordingly.

- “simulation” to “stimulation”

Response: This typo has been corrected accordingly.

Reviewer #3 (Remarks to the Author):

This manuscript ties together observations made over the years by the Yang and Hell labs. It is well written, with well-controlled experiments using state-of-the-art methodology. For the first time I think, we can now see the difference between beta2AR populations phosphorylated by PKA or by GRKs- even though we have known for years that both protein kinases call receptor desensitization. The authors show that GRK mainly phosphorylates a pool of monomeric receptors that internalizes and that PKA phosphorylates a receptor dimer which remains at the plasma membrane as part of a larger signalling complex that contains effector molecules like the L-type calcium channel.

They show that these monomeric and dimeric receptor complexes are spatially segregated using super resolution microscopy combined with single molecule pulldowns. They had already shown that about 50% of the beta2AR was dimeric using this latter technique and validate those results here both in terms of simple spectral overlap and with step photobleaching approaches.

Moving beyond HEK 293 cells, they show distinct distributions in hippocampal neurons with GRK-sensitive receptors in the soma and PKA-sensitive receptors in dendrites- which occurred during neuronal maturation.

Finally, they show that the PKA-sensitive receptors (in neurons with a WT beta1 and beta2AR KO background) are necessary for modulation of L-type calcium channels.

The data are generally convincing and I enjoyed reading this manuscript. My only criticism as such is it would be interesting to have more discussion about how these receptors adopt their fate in the first place. Why are monomeric receptors more sensitive to GRK than PKA and vice versa? Are the populations capable of dynamic exchange? What is it about neuronal maturation that alters this? I am not asking for additional experiments although the authors could be encouraged to manipulate anterograde receptor trafficking with their mutant receptors. I feel that this paper might be seminal at some point and I just want to help them get it right!

Response: We thank the reviewer for the positive note and creative suggestions. Moving beyond the current manuscript, there are many questions

remaining to be addressed. For example, while our data show that ISO does not induce dynamic changes in ratio of β 2AR monomer and dimer, others have reported that some ligands such as ICI-118551 and Carvedilol are capable to induce dynamic changes in beta2AR monomer and dimer ratio (EMBO J, 2009, 28, 3315-3328 and Chem. Commun., 2016, 52, 7086-7089). In the same vein, we think that certain ligands may be able to induce dynamic changes in GRK and PKA phosphorylated populations. At the moment, we do not have a clear idea on why monomeric receptors are more sensitive to GRK than PKA and vice versa. Potential explanations may include: the pre-coupling of different kinase (and scaffold proteins) with receptors; different accessibility of two kinases (receptor intracellular loops vs. C-terminal); and different locations at lipid rafts or non-rafts microdomains.

Meanwhile, we have attempted to examine whether neuronal maturation alters this dynamic. Our initial attempts were failed due to the low efficiency in gene delivery into cultured neurons. Other approaches are needed to overcome the issues before the analysis becomes feasible. Meanwhile, in our preliminary study, removal of GRK phosphorylation sites leads to higher monomer ratio (both at resting state and after agonist stimulation), while removal of PKA phosphorylation sites has no effects on receptor ratio. These are the questions that we look forward to pursuing in the near future.

We have revised the Discussion section to include the above responses.

REVIEWERS' COMMENTS:

Reviewer #1 (Remarks to the Author):

In the revised version, Shen et al addressed all the comments raised by this reviewer. As indicated in my comments to the original manuscript, I find the experiments suited for the biological question, well performed and support the conclusions. I therefore recommend publication.

Minor comments:

1. following my previous comment, graphs has been modified to scatter dot plots. However, for some reason those are overlaid on the previous graphs. I think all graphs should be modified to resemble the graphs presented in Fig 2b and 4c,d.
2. Figure Legend: Figure 1a, "SIM super resolution microscopy" should be changed to "SIM imaging". SIM stands for Structured illumination microscopy.
3. In all figure legend please replace the term "3 repeats" with "3 independent experiments"
4. Figure legend: Figure 1 - it should be clearly stated that the cells are transfected with a FLAG tagged version of the receptor.
5. replace "Flag" with "FLAG".
6. Figure legend: Figure 2a: "Confocal imaging shows total beta2ARs. I could not find the images in the figure. They should be added or the text should be revised.
7. Methods: Structured illumination super-resolution microscope and confocal imaging. It should be indicated that SIM data was reconstructed and the number of rotations should be specified.
8. Discussion: the experiments were done under over expression conditions which changes the stoichiometry of the receptor in the cells and by that can potentially affect the results. this should be clearly indicated in the discussion.

Reviewer #2 (Remarks to the Author):

This is an excellent and important study. The authors have either clarified, added new data, or presented data in an improved format to address previous comments.

Only two minor and related questions/points of clarification:

1. Single channel recording/analysis. The authors added new experimental details in this revision which are helpful. However, it would be helpful to state explicitly that the number of channels in each patch recording (n) was not estimated and that all data are presented as " nPo " (product of n and channel open probability).
2. In the results section (p. 7) and in the legend to figure 5 (p. 14) channel open probability is defined as (nPo). "...abolished ISO-induced increases in open probability (nPo) of LTCC Cav1.2 as measured by single-channel recordings in hippocampal neurons..." "The open probability (nPo) of

LTCC Cav1.2 was quantified from panel..." As P_o is not measured, the authors should refer to changes in overall channel activity (nP_o) consistent with the methods section.

We would like to thank again for all referees for their thorough and helpful feedback to improve the manuscript. A point-by-point response to referees' comments is given below.

Reviewer #1 (Remarks to the Author):

In the revised version, Shen et al addressed all the comments raised by this reviewer. As indicated in my comments to the original manuscript, I find the experiments suited for the biological question, well performed and support the conclusions. I therefore recommend publication.

Minor comments:

1. Following my previous comment, graphs has been modified to scatter dot plots. However, for some reason those are overlaid on the previous graphs. I think all graphs should be modified to resemble the graphs presented in Fig 2b and 4c,d.

Response: We have now used scatter dot plots to show all graphs, and removed bars shown in previous graphs.

2. Figure Legend: Figure 1a, "SIM super resolution microscopy" should be changed to "SIM imaging". SIM stands for Structured illumination microscopy.

Response: We have changed to "SIM imaging" accordingly.

3. In all figure legend please replace the term "3 repeats" with "3 independent experiments"

Response: We have replaced the term "repeats" with "independent experiments".

4. Figure legend: Figure 1 - it should be clearly stated that the cells are transfected with a FLAG tagged version of the receptor.

Response: We have added "cells expressing FLAG-tagged β_2 AR" into the Figure 1 legend.

5. Replace "Flag" with "FLAG".

Response: We have replaced "Flag" with "FLAG".

6. Figure legend: Figure 2a: "Confocal imaging shows total beta2ARs. I could not find the images in the figure. They should be added or the text should be revised.

Response: We thank the referee for pointing out this mistake. We have removed "total β_2 AR" and "anti-FALG" from Figure 2 legend.

7. Methods: Structured illumination super-resolution microscope and confocal imaging. It should be indicated that SIM data was reconstructed and the number of rotations should be specified.

Response: We thank the referee for the suggestion. We have now added the number of rotations and clearly stated that SIM data were reconstructed images (in page 12).

8. Discussion: the experiments were done under over expression conditions which changes the stoichiometry of the receptor in the cells and by that can potentially affect the results. This should be clearly indicated in the discussion.

Response: We thank the referee for the suggestion. We have revised the Discussion section to include the above information (in page 9).

Reviewer #2 (Remarks to the Author):

This is an excellent and important study. The authors have either clarified, added new data, or presented data in an improved format to address previous comments.

Only two minor and related questions/points of clarification:

1. Single channel recording/analysis. The authors added new experimental details in this revision which are helpful. However, it would be helpful to state explicitly that the number of channels in each patch recording (n) was not estimated and that all data are presented as " nPo " (product of n and channel open probability).

Response: We have now added this statement in Methods section (in page 17)

2. In the results section (p. 7) and in the legend to figure 5 (p. 14) channel open probability is defined as (nPo). "...abolished ISO induced increases in open probability (nPo) of LTCC Cav1.2 as measured by single-channel recordings in hippocampal neurons..."

"The open probability (nPo) of LTCC Cav1.2 was quantified from panel..." As Po is not measured, the authors should refer to changes in overall channel activity (nPo) consistent with the methods section.

Response: We thank the referee for the suggestion. We have changed "open probability (nPo)" to "overall channel activity (nPo)".